# TNF-α antagonists differentially induce TGF-β1-dependent resuscitation of dormant-like *Mycobacterium tuberculosis*

**Ainhoa Arbués**[1,2], **Dominique Brees**[3], **Salah-Dine Chibout**[3], **Todd Fox**[4], **Michael Kammüller**[3]*, **Damien Portevin**[1,2]*

**1** Department of Medical Parasitology & Infection Biology, Swiss Tropical and Public Health Institute, Basel, Switzerland, **2** University of Basel, Basel, Switzerland, **3** Novartis Institutes for Biomedical Research, Basel, Switzerland, **4** Novartis Pharma AG, Basel, Switzerland

* michael.kammueller@novartis.com (MK); damien.portevin@swisstph.ch (DP)

**Data Availability Statement:** All relevant data are within the manuscript and its Supporting Information files.

## Abstract

TNF-α- as well as non-TNF-α-targeting biologics are prescribed to treat a variety of immune-mediated inflammatory disorders. The well-documented risk of tuberculosis progression associated with anti-TNF-α treatment highlighted the central role of TNF-α for the maintenance of protective immunity, although the rate of tuberculosis detected among patients varies with the nature of the drug. Using a human, *in-vitro* granuloma model, we reproduce the increased reactivation rate of tuberculosis following exposure to Adalimumab compared to Etanercept, two TNF-α-neutralizing biologics. We show that Adalimumab, because of its bivalence, specifically induces TGF-β1-dependent *Mycobacterium tuberculosis* (*Mtb*) resuscitation which can be prevented by concomitant TGF-β1 neutralization. Moreover, our data suggest an additional role of lymphotoxin-α–neutralized by Etanercept but not Adalimumab–in the control of latent tuberculosis infection. Furthermore, we show that, while Secukinumab, an anti-IL-17A antibody, does not revert *Mtb* dormancy, the anti-IL-12-p40 antibody Ustekinumab and the recombinant IL-1RA Anakinra promote *Mtb* resuscitation, in line with the importance of these pathways in tuberculosis immunity.

## Author summary

*Mycobacterium tuberculosis* (*Mtb*) is the world's leading infectious killer. Multi-cellular immune structures called granulomas may constitute a latent form of *Mtb* infection and a potential reservoir for future cases. Post-marketing surveillance data suggested that *Mtb* protective immunity is unequally impacted by different TNF-α-targeting drugs used to treat inflammatory disorders. We used an *in-vitro* granuloma model to reproduce these clinical observations and gain mechanistic insights and, in addition, to assess the risk of tuberculosis reactivation associated with the use of other immunomodulatory drugs. These results may inspire pharmacologists to design future drug-development strategies of biologics in particular, while immunologists and microbiologists will find a relevant

**Funding:** DP received the funding to conduct the study under a research agreement contract between Novartis AG and the Swiss Tropical and Public Health Institute. The funders initiated study design e.g. decision to implement the assay and compare activity of the tested drugs. The funders later supported further study design initiated by the collaborators, and had no role in data collection and analysis.

**Competing interests:** DP has received a research grant from Novartis. D.B., S.C., T.F. and M.K. are full-time employees of Novartis.

experimental approach to disentangle the complex interactions involved in *Mtb* protective immunity and immunopathogenesis.

## Introduction

Tuberculosis (TB) remains the leading cause of deaths worldwide due to a single infectious agent. In addition, it is estimated that a quarter of the world's population presents an immune memory against *Mycobacterium tuberculosis* (*Mtb*)-specific antigens in the absence of clinical symptoms, and is thus inferred to be latently infected. Therefore, so-defined latent TB infection (LTBI) does not necessarily reflect the presence of a continued *Mtb* infection as it encompasses cured as well quiescent, asymptomatic or subclinical infections [1]. Recent *Mtb* infection in high-transmission areas is the major contributor to the global TB burden [2]. Yet, in low endemic countries, the risk of progressing from latent to active TB can reach up to 10% if the immune system is weakened, e.g. as a consequence of HIV co-infection or immunosuppressive drug treatments.

The hallmark of the host immune response against the tubercle bacillus is the formation of structurally-organized, multicellular clusters constituted mainly of macrophages and lymphocytes called granulomas. Despite having the potential to be sterilizing, in some instances granulomas may contain but not eliminate the infection. Current thinking holds that immune activation and hypoxia within granulomas favor a switching of mycobacterial physiology into a lipid-rich, low-metabolic, and potentially non-replicating, dormant state that may persist for decades. Consequently, dormant *Mtb* displays an increased tolerance to antibiotics that target metabolic pathways active during bacterial replication [3,4]. The complex pathophysiology of *Mtb* infection suscitated the need to define an appropriate terminology. While latency and reactivation respectively refer to absence or presence of clinical symptoms, dormancy and resuscitation describe bacterial phenotypes characterized by repressed or revived levels of replication and metabolic activity, respectively [5,6]. The metabolic switch leading to dormancy or non-replicating persistence can be induced *in vitro* upon exposure to various stresses including hypoxia. Under hypoxic conditions *Mtb* accumulates intracellular triacylglycerides into lipid inclusions, and undergoes transcriptional changes leading to a shift in carbon and energy metabolism [7].

A well-established host factor controlling *Mtb* dormancy is tumor necrosis factor (TNF)-α, as documented by the clinical association of anti-TNF-α therapies with reactivation of LTBI [8]. TNF-α is a homo-trimeric cytokine produced by a variety of immune cells with pleiotropic functions essential for the control of mycobacterial infections [9,10]. It promotes control of *Mtb* intracellular growth within phagocytes [11,12], and also contributes to cell recruitment and consequently, granuloma formation [13]. TNF-α is initially produced as a transmembrane form (tmTNF-α) which can then be released upon specific enzymatic activity mediated by the TNF-α converting enzyme (TACE) [14]. tmTNF-α also plays a role in the inflammatory response signaling either directly into TNF receptor-bearing cells, and also reciprocally transmitting outside-to-inside (reverse) signals into tmTNF-α-expressing cells themselves [15].

Various biological drugs targeting TNF-α are currently used for the treatment of immune-mediated inflammatory disorders. These encompass notably infliximab (IFX), a humanized mouse monoclonal antibody; adalimumab (ADA), a fully-human monoclonal antibody; and etanercept (ETA), a soluble form of the human TNF-α receptor type II (TNFR2) fused to an Fc fragment. The fact that treatment with TNF-α-targeting biologics increases the risk of TB was observed shortly after their licensing 20 years ago [16]. However, post-marketing

surveillance data suggested that treatment with anti-TNF-α antibodies induces higher LTBI reactivation rate in comparison to ETA [17]. A major difference between the two types of TNF-α antagonists resides in their binding properties. On the one hand, antibodies, such as IFX and ADA, bear two binding sites. Consequently, up to three IFX molecules can be bound to a single TNF-α homotrimer and only TNF-α-targeting antibodies can mediate reverse signals through clustering of tmTNF-α [18]. On the other hand, TNFR2, and therefore ETA, can only interact with one single molecule of TNF-α at a time. Moreover, TNFR2 also binds TNF-β (more commonly referred to as lymphotoxin (LT)-α) and, as a consequence, ETA bioactivity may potentially account for the neutralization of both TNFR2 ligands [19].

To date, most studies investigating the immunological mechanisms responsible for the induction of LTBI reactivation have focused on individual TNF-α blockers [20]. Only few authors have performed comparative studies aiming to elucidate the mechanisms behind the differential risk observed between anti-TNF-α antibodies and the receptor fusion protein. Harris *et al*. showed that only IFX and ADA, but not ETA, inhibited the maturation of *Mtb*-containing phagosomes in primary human macrophages [21]. Hamdi and collaborators observed that all three TNF-α-targeting biologics inhibited *Mtb*-specific CD4$^+$ T-cell proliferation from LTBI patients, although ETA was less potent [22]. Finally, the mathematical design of *in-silico* granulomas suggested that differences in drug binding kinetics and vascular permeability could explain the differential rates of TB reactivation associated with the different TNF-α-targeting biologics [23,24].

Several TNF-α- and non-TNF-α-targeting biotherapeutics have expanded the pharmaceutical arsenal for the treatment of immune-mediated inflammatory disorders. The historical concern arising from the post-marketing surveillance of TNF-α antagonists has brought justified cautiousness concerning potentially impaired protective immune responses against *Mtb* infection by these novel biotherapeutics [25]. Hence, the development of tools able to predict TB infection risk in patients treated with biologics could substantially impact the clinical management of the respective disease and as such, benefit both physicians and patients, as well as contribute to a refined understanding of TB protective immunity.

Over the last decade, several independent laboratories described *in-vitro* granuloma models that reflect the organization of human nascent granulomas orchestrated by relevant cytokines, and that constitute valuable tools to study key aspects of the interaction of *Mtb* with the host immune response [26–30]. To further investigate the differential rate of *Mtb* resuscitation observed between TNF-α-inhibitors ADA and ETA, we made use of the three-dimensional model developed by Kapoor and collaborators and that displayed interesting features of *Mtb* associated with dormancy, such as the accumulation of lipid bodies, loss of acid-fastness and an increase in antibiotic tolerance [4,28]. Noteworthy, this model was also able to reproduce the induction of *Mtb* resuscitation upon exposure to a research grade TNF-α-neutralizing antibody.

In this report, we explore the capability of such human, *in-vitro* granuloma model to assess the LTBI-reactivation risk of several TNF-α- and non-TNF-α-targeting biologics licensed for the treatment of various immune-mediated inflammatory disorders. We demonstrate the relevance of this approach by reproducing a differential rate of *Mtb* resuscitation in *in-vitro* granulomas exposed to ADA in comparison to ETA. Finally, we show that the different LTBI-reactivation rate observed for these two TNF-α-targeting biologics arises from divergent mechanisms of action: ADA mediates substantial *Mtb* resuscitation in a TGF-β1-dependent manner via tmTNF-α reverse signaling, while ETA potentiates a mild resuscitation of *Mtb* only through neutralization of TNF-α and, to an unexpected similar extent, LT-α.

## Results

### Human, *in-vitro* granulomas mimic dormant-like *Mtb* features

First of all, we sought to confirm that upon formation of 3D microgranulomas *in vitro*, *Mtb* can exhibit dormancy characteristics such as alteration of gene regulation coupled with the accumulation of triacylglycerides as intracellular lipid inclusions and loss of acid fastness, as studied in more detail by the group of Kolattukudy [28]. Peripheral Blood Mononuclear Cells (PBMCs) isolated from consenting, anonymous healthy blood donors were used. In order to be able to study the impact of biologics targeting cytokines derived from innate as well as adaptive immune responses in this model, only samples displaying significant IFN-γ+ CD4+ T cell responses against *Mtb* protein purified derivative (PPD) were included in the study (S1 Fig). None of the donors displayed signs of LTBI, i.e. a significant response against a synthetic overlapping peptide pool covering the sequences of ESAT-6, CFP-10 and two highly immunogenic peptides of TB7.7 *Mtb* proteins. This suggests that responses to PPD could be attributed to the cross-reactivity of antigens delivered through previous *M. bovis* BCG vaccination and/or to previous exposure to environmental mycobacteria [31–33]. PBMCs were infected with *Mtb* H37Rv and embedded in a matrix of collagen and fibronectin. A representative view of *Mtb*-induced granulomas obtained 8 days post-infection is depicted in Fig 1A. However, and as previously observed [30], granulomas differed in number and size across donors. We used live cell-tracker dyes to show that these structures consist of monocyte-derived macrophages (orange) surrounded by T cells (in green) as well as additional unlabeled mononuclear-cell subsets recalling the organizational features of granulomas *in vivo* (Fig 1B), as previously reported for this model [28]. The formation of *in-vitro* granulomas promoted a significant accumulation of *Mtb* harboring a dormant-like phenotype. Compared to *Mtb* recovered from day 1 post-infection, the proportion of lipid-rich (Nile red-positive) *Mtb* increased 8 days post-infection (Fig 1C). Furthermore, the enriched Nile red-positive *Mtb* phenotype correlated with increased transcription levels of *icl* (isocitrate lyase) and *gltA1* (methylcitrate synthase) and down-regulation of *nuoB* (NADH dehydrogenase, chain B) and *ctaD* (cytochrome c oxidase polypeptide I) (Fig 1D), as reported previously by Kapoor *et al* [28], and corresponding to transcriptome profiles characteristic of lipid-rich persister-like bacilli found in clinical tuberculous sputum [34].

### Human, *in-vitro* granulomas corroborate LTBI reactivation related to treatment with TNF-α- and some non-TNF-α-targeting biologics

Next, we assessed the capability of some TNF-α- and non-TNF-α-targeting biotherapeutics to potentially impact *Mtb* dormancy in this *in-vitro* granuloma model. We focused our investigation on cytokine antagonists already licensed for the treatment of immune-mediated inflammatory disorders. These encompass ADA, an anti-TNF-α antibody; ETA, a chimeric human TNFR2 fused to an immunoglobulin Fc fragment; ustekinumab (UST), a human monoclonal antibody targeting the IL-12p40 subunit, a constituent of both IL-12 and IL-23; anakinra (ANA), a recombinant, non-glycosylated version of the human IL-1 receptor antagonist (IL-1RA); and secukinumab (SEK), a human monoclonal anti-IL-17A antibody. Four days post-infection, nascent granulomas were exposed to equimolar concentrations of each compound individually, or a human IgG1 isotypic control (Iso). No obvious morphological differences could be detected between granulomas treated with any of the investigated cytokine antagonists compared to the isotype control. Nonetheless, each individual compound differentially interfered with the capacity of granulomas to maintain *Mtb* in a dormant-like state (Fig 2A). ADA induced the highest level of *Mtb* resuscitation, using as proxy an increase in the

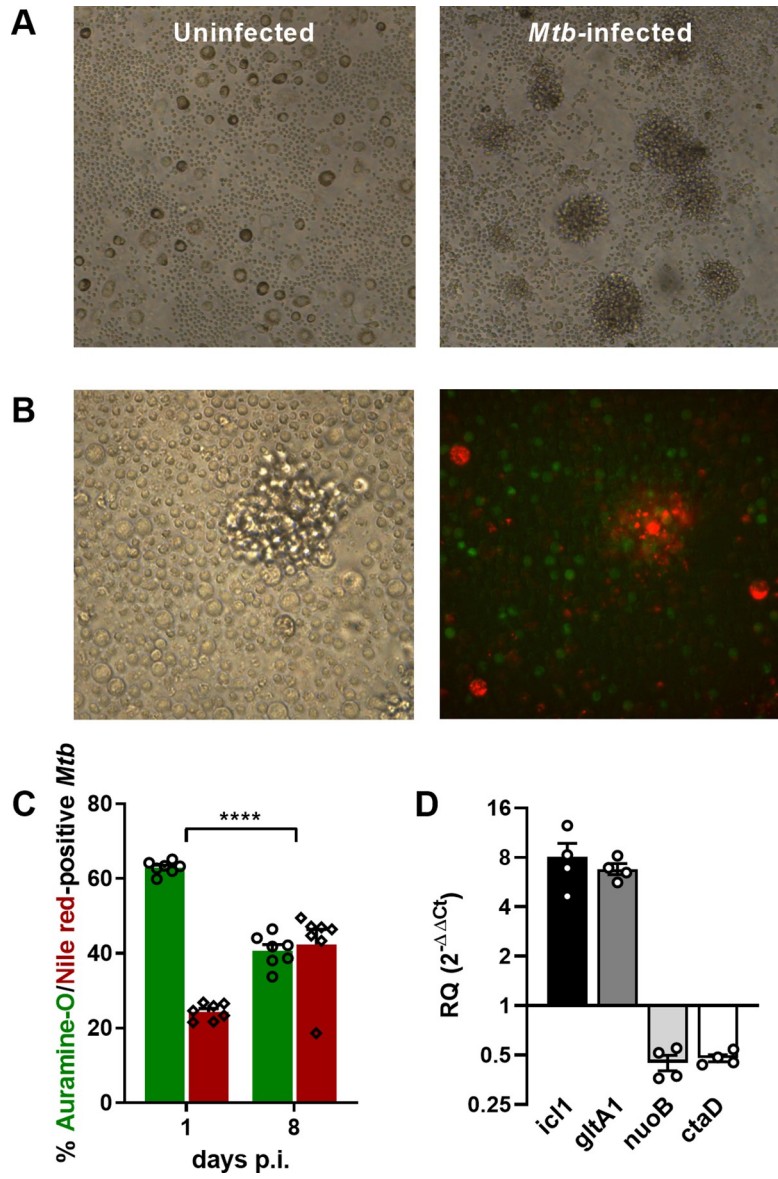

**Fig 1. Human, *in-vitro* granulomas mimic dormant-like *Mtb* features.** **(A)** Representative bright-field microscopy pictures of 3D *in-vitro* granulomas formed 8 days post-infection with *Mtb* H37Rv compared to uninfected PBMCs. **(B)** Representative structure of *in-vitro* granulomas under bright-field (left panel) and fluorescence microscopy (right panel). Monocytes/macrophages were labeled in orange and CD4$^+$ T cells in green. **(C)** Percentages of auramine-O-(green) and Nile red-positive (red) *Mtb* quantified by fluorescence microscopy (mean ± SEM from 7 independent donors) before (1 day post-infection) or after (8 days post-infection) granuloma formation. Statistical analysis was performed using a generalized linear mixed-effects model; ****, p<0.0001. **(D)** Relative expression values for *icl1*, *gltA1*, *nuoB* and *ctaD* after granuloma formation (8 days post-infection) determined by qRT-PCR (mean ± SEM from 4 independent donors). Results are expressed as fold change in log2 scale relative to an aerobically-grown, mid-log *Mtb* culture, using 16S rRNA as the endogenous control.

representation of metabolically-active (auramine-O-positive) bacteria and a concomitant decrease in the percentage of dormant-like (Nile red-positive) *Mtb*, compared to the isotype control. UST and ANA also promoted a marked, comparable reduction in the frequency of dormant-like bacteria. Interestingly, ETA, despite sharing the same target with ADA, induced an intermediate level of *Mtb* resuscitation. SEK behaved as the isotype control, displaying no

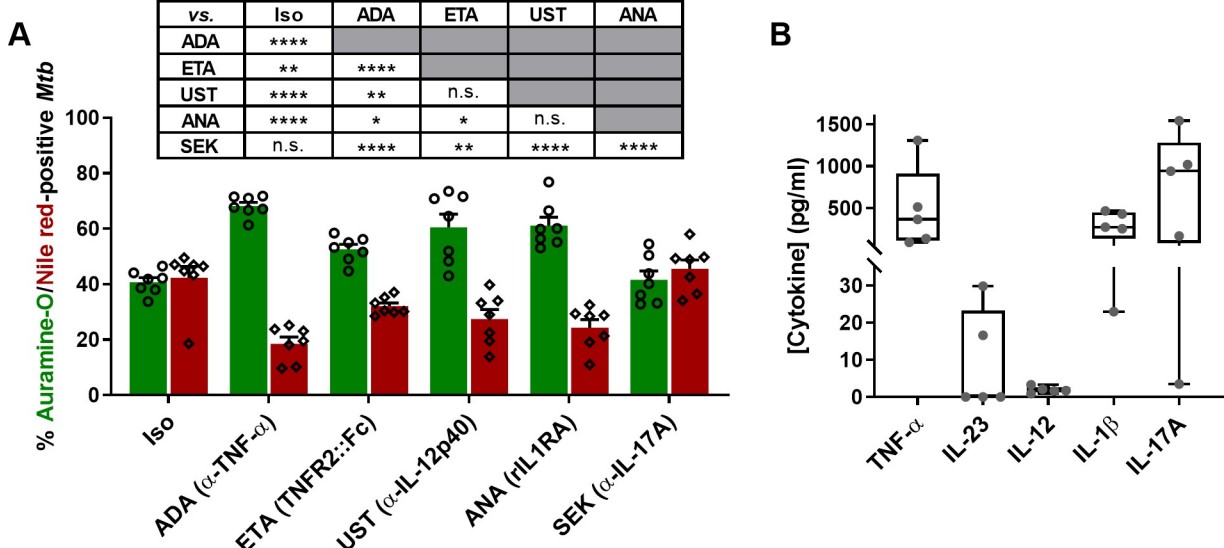

**Fig 2. Human, *in-vitro* granulomas corroborate risk of LTBI reactivation linked to treatment with TNF-α-targeting biologics. (A)** Percentages of auramine-O- (green) and Nile red-positive (red) *Mtb* quantified by fluorescence microscopy (mean ± SEM from 7 independent donors) following 4 days of exposure with an isotype control (Iso), adalimumab (ADA), etanercept (ETA), ustekinumab (UST), anakinra (ANA) or secukinumab (SEK). Statistical analysis was performed using a generalized linear mixed-effects model; n.s., not significant; *, p<0.05; **, p<0.01; ****, p<0.0001. **(B)** Cytokine levels measured in supernatants of untreated *in-vitro* granulomas 8 days post-infection (median with interquartile ranges, minimum and maximum values for 5 independent donors).

reversal of the mycobacterial dormant-like phenotype into an active state, confirming independently the results reported previously [35]. A minor but constant proportion of bacteria simultaneously stained with both auramine-O and Nile red could be detected in all treatment groups, likely representing transitional states.

Next, and notably given the lack of activity displayed by SEK, we assessed if the cytokines targeted by the investigated biologics were being actively produced upon granuloma formation. Untreated *in-vitro* granulomas showed the following cytokine levels–median (25th–75th percentiles)–in the supernatant on day 8 post-infection (Fig 2B): TNF-α 367.8 pg/ml (115.6–912.1 pg/ml); IL-1β 274.4 pg/ml (138.8–450.4 pg/ml); unexpectedly, and contrasting the major increase in the proportion of metabolically-active *Mtb* recovered from granulomas treated with UST, only low levels of IL-23 (0 pg/ml; 0–23.23 pg/ml) or IL-12p70 (1.713 pg/ml; 1.28–2.698 pg/ml) could be detected; and, despite the lack of *Mtb* resuscitation in granulomas exposed to SEK, high amounts of IL-17A (942.8 pg/ml; 85.47–1281 pg/ml) were secreted by most of the donors. Taken together, these results demonstrate the capacity of this human, *in-vitro* granuloma model to identify the capability of TNF-α- and non-TNF-α-targeting biologics to differentially impact *Mtb* dormancy in a manner consistent with preclinical and clinical observations.

## Human, *in-vitro* granulomas reproduce the differential risk of LTBI reactivation clinically observed with ADA or ETA

TNF-α plays a key role in the control of *Mtb* infection, yet the incidence of TB is higher in patients receiving ADA compared to those receiving ETA [36,37]. Our results obtained from granulomas exposed to these two biologics also suggested a preferential resuscitation of *Mtb* in the context of ADA treatment. Consequently, we sought to decipher the mechanisms underlying the differential interference of ADA and ETA with granuloma-induced *Mtb* dormancy in

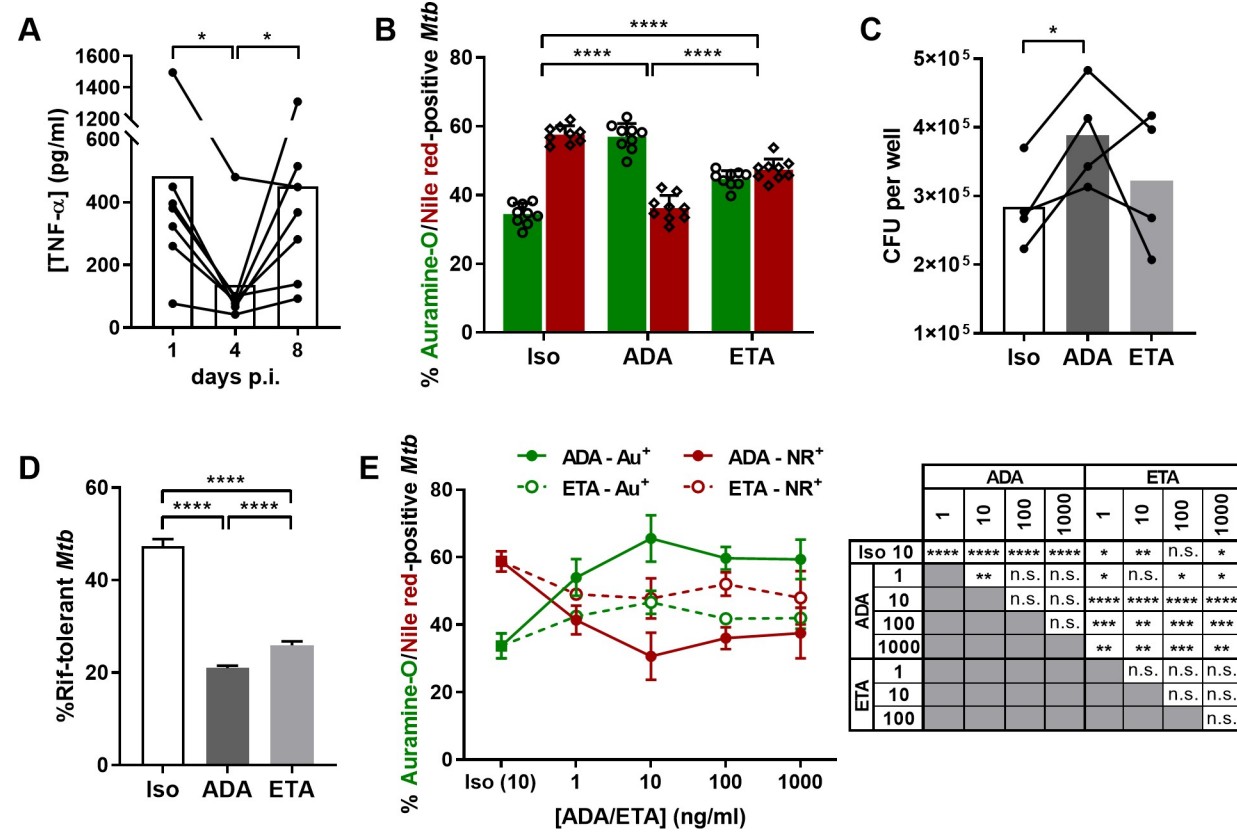

**Fig 3. Unequal interference of TNF-α-targeting biologics ADA and ETA with granuloma-induced *Mtb* dormancy does not relate to differential bioavailability. (A)** TNF-α concentration in supernatants on days 1, 4 and 8 post-infection with *Mtb* H37Rv (values for 7 independent donors are represented by line-connected circles and bars picture the mean concentration). Statistical analysis was performed using a Wilcoxon paired test. **(B-D)** At day 4 post-infection, granulomas were exposed for four additional days to an isotype control (Iso), adalimumab (ADA), or etanercept (ETA) at 10 ng/ml. **(B)** Percentages of auramine-O- (green) and Nile red-positive (red) *Mtb* quantified by fluorescence microscopy (mean ± SEM from 9 independent donors). **(C)** *Mtb* bacterial load quantified by CFU assessment (values for 4 independent donors are represented by line-connected circles and bars picture the mean concentration). Statistical analysis was performed using a paired t test. **(D)** Percentage of rifampicin (Rif)-tolerant *Mtb* quantified by CFU (mean ± SEM from 2 independent donors). **(E)** Percentages of auramine-O- (green) and Nile red-positive (red) *Mtb* quantified by fluorescence microscopy following four days of exposure to Iso (full squares), ADA (full circles/continuous line), or ETA (open circles/dotted line) at the indicated concentrations (mean ± SEM from 2 independent donors). Unless stated differently, statistical analysis was performed using a generalized linear mixed-effects model; n.s., not significant; *, p<0.05; **, p<0.01; ***, p<0.001; ****, p<0.0001.

an independent set of experiments. First, we characterized the kinetics of TNF-α accumulation in order to assess the appropriateness of the antagonist treatment timing. The concentration of TNF-α detected alongside the formation of untreated granulomas is depicted in Fig 3A. We observed that the secretion of TNF-α occurs in two waves: an early secretion is detected within the first 24h of infection which significantly wanes during the following days. A second wave of TNF-α accumulates between 4 and 8 days post-infection, concurring with the duration of the ADA and ETA treatment. From an additional set of donors, we confirmed that the presence of ADA significantly reverts dormant-like *Mtb* into metabolically-active (auramine-O-positive) bacilli (Fig 3B). Compared to results depicted in Fig 2A, the effect of ETA appeared less pronounced, yet remained always statistically significant when compared to the Iso- or ADA-treated samples. This appeared to be the consequence of a decreased baseline ratio of auramine/Nile red-positive bacteria induced upon granuloma formation in the absence of drugs (Iso). Complementary results defining dormant-like *Mtb* phenotype were obtained by

measuring bacterial load (Fig 3C) and tolerance to rifampicin (Rif) (Fig 3D). Four days after *Mtb* infection, *in-vitro* granulomas were exposed to ADA, ETA or the isotype control for four additional days. In order to assess the percentage of Rif-tolerant *Mtb*, granulomas were treated, or not, with 5 μg/ml Rif for three extra days followed by determination of CFU. As shown in Fig 3B, exposure to ADA induced significant *Mtb* resuscitation reflected by a significantly higher mycobacterial load (Fig 3C) and a lower percentage of Rif-tolerant bacilli when compared to the isotype (Fig 3D). On the other hand, granulomas treated with ETA showed a slightly lesser (although statistically significant) decrease in the load of Rif-tolerant *Mtb* (Fig 3D), but did not undergo significant changes in their bacterial burden (Fig 3C). Taken together these results confirm again that, in accordance with clinical observations, ETA interferes significantly less with granuloma-induced *Mtb* dormancy than ADA. Differences in the pharmacokinetic properties and bioavailability between these two TNF-α antagonists have been proposed to account for this different propensity. To test this hypothesis, granulomas were exposed to increasing concentrations of ADA or ETA (1, 10, 100, and 1000 ng/ml) prior to the assessment of *Mtb* auramine-O/Nile red phenotypes. Our results reveal a dose-response effect of ADA and ETA on the frequency of metabolically active bacteria between 1 and 10 ng/ml, reaching a plateau beyond 10 ng/ml (Fig 3E). Even a 100-fold higher concentration of ETA was not able to reach the level of *Mtb* resuscitation induced by ADA, suggesting that other mechanisms are responsible for the differential rate of *Mtb* resuscitation in granuloma exposed to ADA in comparison to ETA.

## ADA specifically mediates TGF-β1-dependent resuscitation of dormant-like *Mtb* within granulomas

We investigated whether other intrinsic differences between ADA and ETA could account for their differential bioactivity. As shown for IFX, ADA could interact with tmTNF-α and trigger reverse signaling and subsequent production of TGF-β1 by macrophages whereas ETA failed to do so [38]. Consequently, we sought to decipher wether a selective induction of TGF-β1 could explain the increased propensity of ADA to revert granuloma-induced *Mtb* dormancy compared to ETA. We first aimed to elucidate the nature of the cells potentially experiencing tmTNF-α-mediated reverse signaling. Having selectively used CD4[+] T-cell responders to PPD in this study, we intuitively expected that T cells would be mainly responsible of the late wave of TNF-α observed following granuloma formation. After an overnight incubation with brefeldin A, uninfected or *Mtb*-infected PBMCs were released from the extracellular matrix at the indicated time-points and analyzed by flow cytometry (S2 Fig). As shown in Fig 4, all investigated cell types, encompassing macrophages (Fig 4A) as well as CD4[+] (Fig 4B) and CD8[+] T cells (Fig 4C), were found to produce TNF-α 8 days post-infection.

Next, we quantified the induction of active-TGF-β1 in supernatants of granulomas exposed to ADA, ETA or an isotype control (Fig 5A). A consistent, though very mild, accumulation of active TGF-β1 was observed in ADA-treated wells compared to ETA or the isotype control. We consequently aimed to address if such a concentration of TGF-β1 could interfere with granuloma-induced *Mtb* dormancy and lead to mycobacterial resuscitation. Four days post-infection, granulomas were treated with increasing concentrations of recombinant TGF-β1, in combination or not with a TGF-β1-neutralizing antibody, and *Mtb* was recovered four days later. As depicted in Fig 5B, the presence of exogenous TGF-β1 at 8 pg/ml was already sufficient to significantly impact *Mtb* dormancy. In fact, *Mtb* resuscitation was maximal and reached a plateau from 40 pg/ml of exogenous TGF-β1. In addition, the presence of a TGF-β1-neutralizing antibody was able to prevent *Mtb* resuscitation for all TGF-β1 concentrations tested. Taken together, even a slight increase within the picogram range suffices to significantly

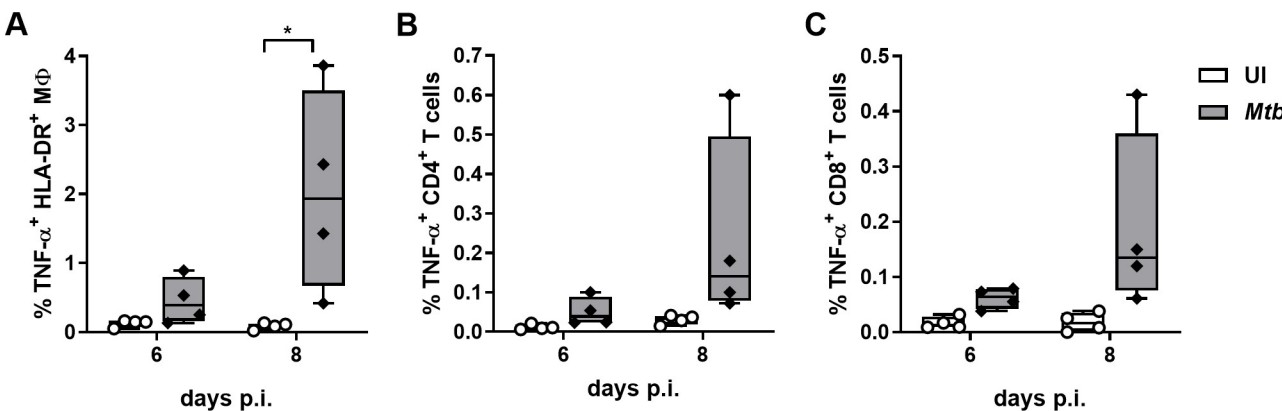

**Fig 4. Late TNF-α production by *Mtb*-induced granulomas originates from T cells as well as macrophages.** Frequencies of TNF-α-producing macrophages (**A**) and CD4+ (**B**) or CD8+ T cells (**C**) from uninfected PBMCs (UI) or granulomas 6 and 8 days post-infection with *Mtb* H37Rv (median with interquartile ranges, minimum and maximum values for 4 independent donors). Statistical analysis was performed using Friedman test; *, p<0.05.

interfere with *Mtb* dormancy within granulomas. To confirm that the enhanced rate of *Mtb* resuscitation observed in granulomas exposed to ADA compared to ETA was indeed due to the selective induction of TGF-β1, we investigated if the addition of a TGF-β1-neutralizing antibody may specifically counteract ADA-related *Mtb* resuscitation. As shown in Fig 5C, neutralization of TGF-β1 in granulomas exposed to ADA completely prevented *Mtb* resuscitation observed in the presence of ADA alone. This phenomenon was found restricted to ADA as the addition of the TGF-β1-neutralizing antibody did not impact the resuscitation of *Mtb* observed in granulomas exposed to ETA. Finally, we tested whether the preferential resuscitation of *Mtb* mediated by TGF-β1 in the presence of ADA may be due to its bivalence and, consequently, to its selective capacity to promote reverse signaling via tmTNF-α cross-linking. To do so, we generated Fab fragments of ADA (ADA-Fab) using immobilized papain (S3 Fig) and compared *Mtb* resuscitation rates within granulomas exposed to ADA or ADA-Fab (concentration normalized according to total TNF-α binding sites) in the presence or absence of the TGF-β1-blocking antibody (Fig 5D). Interestingly, the rate of dormant-like *Mtb* in ADA-Fab-treated granulomas was significantly higher than in granulomas exposed to ADA and, in fact, comparable to granulomas treated with ETA. Confirming our observations depicted in Fig 5C, neutralization of TGF-β1 almost completely prevented the resuscitation rate of *Mtb* associated with ADA. In contrast, and as observed in the case of ETA treatment, the presence of TGF-β1-blocking antibodies did not impact the frequency of auramine-O/Nile red-positive bacteria in granulomas exposed to ADA-Fab (Fig 5D). This observation demonstrates that the bivalence of ADA is required to specifically induce TGF-β1-dependent resuscitation of *Mtb* within granulomas.

## Neutralization of ETA ligand LT-α leads to mild *Mtb* resuscitation

Another intrinsic difference between ADA and ETA is that only ETA can interact with TNF-α as well as TNF-β, also known as LT-α [19]. Therefore, we aimed to elucidate the role of LT-α in the regulation of *Mtb* dormancy in granulomas exposed to ETA. We first assessed the kinetics of LT-α production upon formation of *Mtb* granulomas. As represented in Fig 6A, and in contrast to TNF-α, LT-α only accumulated between 4 and 8 days post-infection. We then interrogated the cellular source of LT-α production following formation of *Mtb* granulomas. After an overnight incubation with brefeldin A, uninfected or *Mtb*-infected PBMCs were

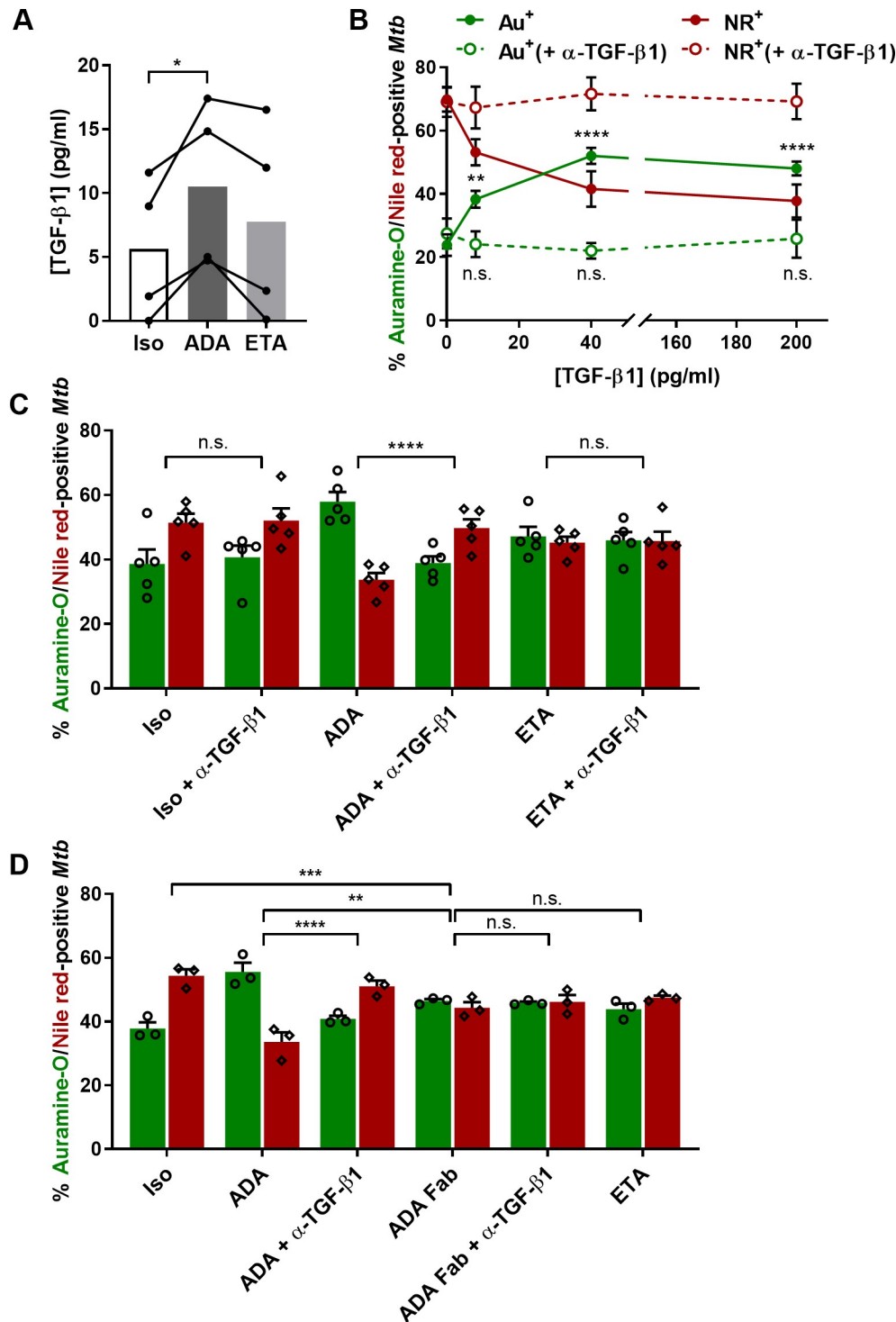

**Fig 5. ADA specifically mediates TGF-β1-dependent *Mtb* resuscitation. (A)** Active TGF-β1 concentration in supernatants of granulomas 8 days post-infection with *Mtb* H37Rv and after 4 days of exposure to either adalimumab (ADA), etanercept (ETA) or an isotype control (Iso) (values for 4 independent donors are represented by line-connected circles and bars depict mean concentration). Statistical analysis was performed using Friedman test. **(B-D)** Averaged percentages of auramine-O- (green) and Nile red-positive (red) *Mtb* quantified by fluorescence microscopy after 4 days of exposure to: **(B)** 0, 8, 40 or 200 pg/ml of recombinant TGF-β1 in the absence (full circles/continuous line) or presence (open circles/dotted line) of a TGF-β1-blocking antibody (mean ± SEM from 3 independent donors); **(C)** an isotype control (Iso), adalimumab (ADA), or etanercept (ETA) in the absence or presence of a TGF-

β1-neutralizing antibody (+ α-TGF-β1) (mean ± SEM from 5 independent donors); and **(D)** ADA or ADA Fab fragment (ADA-Fab) in the absence or presence of a TGF-β1-neutralizing antibody, ETA or Iso. Statistical analysis was performed using a generalized linear mixed-effects model; n.s., not significant; *, p<0.05, **, p<0.01; ***, p<0.001; ****, p<0.0001. For **(B)** all comparisons were performed against the untreated control. For **(C-D)** only the most relevant comparisons were plotted for clarity reasons but results from all combinations are available on S2 Table, panels A and B respectively.

released from the extracellular matrix at the indicated time-points and analyzed by flow cytometry (S2 Fig). We observed that both CD4⁺ (Fig 6B) and CD8⁺ (Fig 6C) T cells produce LT-α in response to *Mtb* infection at 6 and 8 days post-infection. Finally, we investigated the impact of LT-α neutralization on the the rate of dormant-like *Mtb* within granulomas. In order to avoid potential side-effects originating from the bivalence of the antibody, we generated Fab fragments of a specific anti-LT-α neutralizing antibody (α-LTα-Fab) using

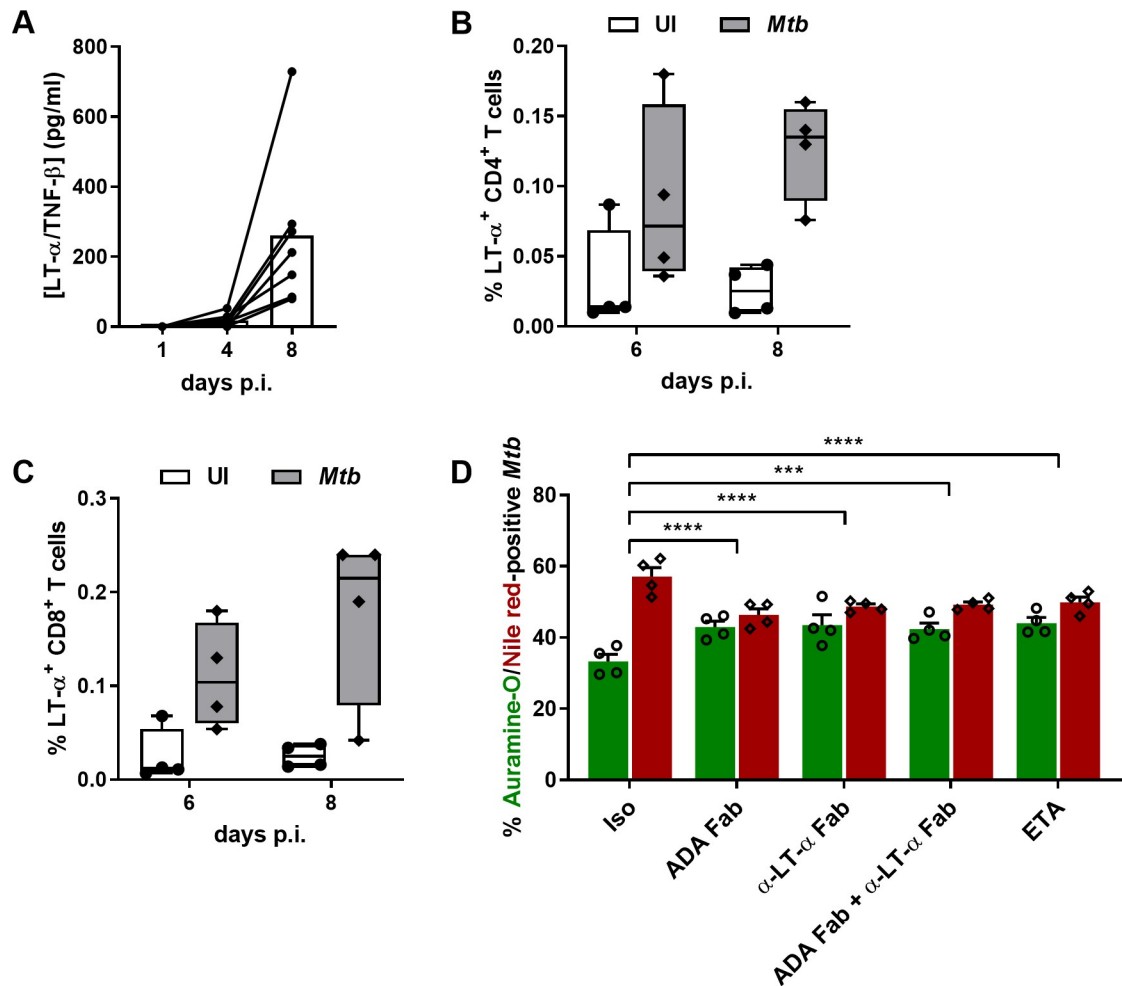

**Fig 6. ETA-specific interference with CD4⁺ and CD8⁺ T cell-derived LT-α sustains mild *Mtb* resuscitation. (A)** LT-α concentration in supernatants on days 1, 4 and 8 post-infection with *Mtb* H37Rv (values for 7 independent donors are represented by lined-connected circles and bars depict mean concentration). **(B-C)** Frequencies of LT-α-producing CD4⁺ **(B)** and CD8⁺ T cells **(C)** from uninfected PBMCs (UI) or granulomas 6 or 8 days post-infection with *Mtb* H37Rv (median with interquartile ranges, minimum and maximum values from 4 independent donors). **(D)** Percentages of auramine-O- (green) and Nile red-positive (red) *Mtb* quantified by fluorescence microscopy (mean ± SEM from 4 independent donors) after four days of exposure to either an isotype control (Iso), etanercept (ETA) or the Fab fragments from adalimumab (ADA-Fab) or an anti-LT-α antibody (α-LTα-Fab). Statistical analysis was performed using a generalized linear mixed-effects model; ***, p<0.001; ****, p<0.0001.

immobilized papain (S3 Fig). We then compared the *Mtb* resuscitation rate within granulomas exposed to ADA-Fab or α-LTα-Fab individually and in combination. Both Fab fragments showed comparable activity to ETA (Fig 6D), pointing to an unexpected capacity of LT-α to promote the development of dormant-like *Mtb* in granulomas. Interestingly, the combined treatment of granulomas with ADA-Fab and α-LTα-Fab did not reveal an additive effect on *Mtb* resuscitation (Fig 6D).

## Discussion

Investigating the complex dynamic interplay between the host and the intracellular pathogen *Mtb* has proven to be challenging. In particular, defining the conditions leading to reactivation from LTBI has been the subject of numerous studies in various animal species and humans [39]. The importance of CD4+ T cells, TNF-α, IFN-γ, IL-12p40, together with the IL-1/IL-1R1 pathway, in host resistance to intracellular *Mtb* infection is evident from animal models and human inherited and acquired immunodeficiencies [40]. Still many questions remain unanswered concerning the importance of other host immunological factors for the control of LTBI. In recent years, novel pathway-specific biotherapeutics leading to selective immunosuppression have become available for the treatment of inflammatory immune-mediated diseases [41,42]. Logically, this partial immunosuppression has proven to come at the expense of an increased susceptibility to particular viral, fungal or bacterial infections and triggered inquiries into the significance of these immunological pathways concerning LTBI reactivation [43,44]. Eventually, only few cases of TB are being detected among patients treated with non-TNF-α-targeting biologics raising ethical concerns on the relevance of LTBI pre-screening requirement and prophylactic antibiotic treatment with potentially hepatotoxic drugs [25,45]. However, clinical trials assessing these new compounds either excluded participants displaying signs of LTBI, or compound labels systematically recommended LTBI screening and prophylactic antibiotic therapy prior treatment initiation [46].

In search for a translational model mimicking the dynamic host-pathogen interplay in TB, we focused on *in-vitro Mtb* dormancy and resuscitation as a preclinical surrogate model of clinical latency and reactivation [5, 6]. Human, *in-vitro* granulomas induce several dormant-like *Mtb* features, such as accumulation of triacylglycerides as intracellular lipid inclusions and loss of acid fastness [28]. In this report, we present a compilation of evidence supporting the relevance of granuloma-like structures induced upon *Mtb* infection of PBMCs from pre-immune blood donors to corroborate the risk of TB infection associated with the use of TNF-α- and some non-TNF-α-targeting biologics. Moreover, we showed that the *in-vitro* granuloma model constitutes a powerful tool to perform mechanistic investigations to dissect the interaction between biologics and granuloma functionality.

Indeed, the conclusions presented here advanced our understanding of the underlying mechanisms supporting the differential rate of LTBI reactivation in patients treated with ADA and ETA despite targeting the same cytokine. In accordance with clinically available data, ADA showed increased propensity for *Mtb* resuscitation when compared to ETA in *in-vitro* granulomas. A computational model suggested that differences in the permeability and therefore diffusion into TB lesions could be responsible of this phenomenon [24]. However, our data revealed that a 100-fold increase in the concentration of ETA could not match the level of ADA-induced *Mtb* resuscitation suggesting that another mechanism is likely involved to explain this differential activity. Our results demonstrate that a specific induction of TGF-β1 is responsible for the increased rate of *Mtb* resuscitation concomitant with ADA treatment compared to ETA. This finding is consistent with the fact that macrophage-derived TGF-β1 would play a major role in TB immuno-pathogenesis [47] and that only antibodies can crosslink

tmTNF-α and trigger reverse signaling leading to TGF-β1 production [38]. Additional RNA interference experiments could attest if this activity relies on *de-novo* TGF-β1 production or an increased conversion of the latent forms present in human serum. However, transfection of *in-vitro* granulomas embedded in a matrix of collagen would be particularly challenging. Our observations are also consistent with the fact that addition of exogenous TGF-β1 accelerates *Mtb* replication in monocytes [48], while treatment with neutralizing antibodies or natural inhibitors augments their capacity to control *Mtb* growth [49]. Complement-mediated lysis of effector T cells that expressed surface TNF-α, described in rheumatoid arthritis and ankylosing spondylitis patients [50], is unlikely playing a role in our model for our protocol uses de-complemented human serum. Nonetheless, a reduction in effector T cells due to intrinsic TGF-β1 produced by tmTNF-α reverse signaling could add-up to regulation of macrophage functions mediated by TGF-β1 and contribute to the preferential *Mtb* resuscitation in patients treated with ADA compared to ETA. In line with this, and using an *in-silico* granuloma model, Warsinske and collaborators proposed that the presence of TGF-β1 in granulomas inhibits killing of infected macrophages by cytotoxic T cells [51]. Hence, concomitant neutralization of TGF-β1 in patients under ADA therapy could be used to decrease the TB risk associated with this treatment. Furthermore, since we found that TGF-β1 induction is directly linked to ADA bivalence, monovalent or bispecific neutralizing agents could constitute a safer option to prevent the undesired induction of tmTNF-α reverse signaling.

Rather unexpectedly, we found that the specific neutralization of LT-α in the environment of granulomas interfered with *Mtb* dormancy to the same extent than neutralization of TNF-α mediated by ADA Fab fragments. We could not detect an additive activity of blocking both TNF-α and LT-α. This may reflect that the receptor could still interact with one or the other cytokine while bound to a Fab entity. As such, this may compete with the binding of the other cytokine or indirectly prevent its signaling after internalization of the receptor. However, our observations suggest that LT-α may play a stronger role than previously expected in the immune function of granulomas and more specifically also contribute to the mild LTBI reactivation risk associated to ETA. Indeed, the high susceptibility of TNF-α knock-out mice to *Mtb* infection, despite expressing normal levels LT-α, led to the conclusion that LT-α was not required for the immune response against mycobacteria [52]. Nonetheless, in the context of BCG infection, reintroduction of a functional copy of the LT-α gene in TNF-α/LT-α double deficient mice improved their survival [53]. Adding to the controversy, deficient chimeric mice rather suggested an important role of soluble LT-α in the control of *Mtb* infection [54], while the construction of LT-α knock-out mice able to produce normal levels of TNF-α pointed to a minor role of LT-α in the control of chronic TB compared to the major role of TNF-α in the control of acute *Mtb* infection [55].

Ultimately, we showed that granulomas exposed to biologics neutralizing cytokines potentially important for the control of *Mtb* infection could variably impact the physiology of the bacteria reflecting different potential to promote LTBI reactivation. Indeed, individual compounds reproducibly showed variable activity on *Mtb* dormancy, ranging from none (SEK) to moderate (ETA) and more active (ADA, UST and ANA). Regarding the activity linked to ADA and ETA, it is well established that TNF-α plays a critical role in the control of *Mtb* proliferation and granuloma formation [9–13]. Thus, unsurprisingly, the activity of ADA and ETA observed in human *in-vitro* granulomas confirmed the observed risk of LTBI reactivation concurrent to their usage in the clinic. The high activity of UST is in agreement with the previously described natural susceptibility to mycobacterial infections of humans carrying mutations in the IL-12 pathway, activation of which is an important trigger of classical activation of macrophages and induction and maintenance of protective IFN-γ-producing CD4$^+$ T cells [56, 57]. Although not as frequent as with ADA, cases of TB have been reported following UST

therapy [45,58,59]. Despite barely detectable levels of IL-12p70 and IL-23, UST clearly reverts *Mtb* dormancy (Fig 2A), demonstrating that the system is able to capture the consequences of neutralizing a cytokine that does not accumulate. This is reminiscent of IL-10 which plays and important role in controlling *Mtb* but may be challenging to detect due to low expression and inherent instability [60]. IL-1RA, and hence ANA, binds non-productively the IL-1 receptor inhibiting the activities of both IL-1α and IL-1β which are essential for the control of *Mtb* infection in mice [61,62]. The activity observed with ANA is therefore expected and actually consistent with a case report of TB reactivation observed in a rheumatoid arthritis patient receiving this compound [63]. It is also supported by the proposed beneficial role of IL-1R1 signaling during TB infection that counteracts in a PGE2-dependent manner a detrimental production of type I interferons [64]. Finally, the role of IL-17A, in the immune response during TB remains controversial. Knock-out mice for IL-17RA appeared more susceptible to a high-dose intra-tracheal infection with *Mtb* [65], whereas no differences in bacterial burden were observed after a low-dose aerosol infection [66]. Despite IL-17A being actively released upon *in-vitro* granuloma formation, the presence of SEK did not trigger *Mtb* resuscitation within granulomas, supporting independently a previous report [35]. As reviewed recently [67], PBMC-based, *in-vitro* granuloma models usually lack neutrophils, non-hematopoietic-derived cells, vascularization, plasticity, and continuous influx of freshly recruited immune cells, which constitute important limitations. Since not all cellular targets of IL-17A (e.g. neutrophils) are present in the *Mtb*-induced, *in-vitro* granulomas, a potential *in-vivo* effect of its blocking cannot be completely ruled out. Nonetheless, a side-by-side comparison of the effects of anti-IL-17A or anti-TNF-α neutralizing antibodies in a murine infection model, confirmed the importance of TNF-α in the immune response against *Mtb*, in contrast to the IL-17 pathway [68]. Altogether, to date, the composite of clinical, *in-vivo* and *in-vitro* data show a low risk for mycobacterial infection under SEK therapy, in contrast to anti-TNF-α treatment [45].

From a translational safety assessment perspective, identification of drug-induced hazards or infection risks has proven to be challenging for a number of reasons. First, the precise nature of immune responses and built-in reserve capacity keeping commensal or pathogenic microbes in check is not comprehensively understood. Secondly, time-dependent contributions of specific cytokines by polyfunctional immunocompetent cells are essential to multicellular host responses in a protective immunity network [69], and hence complicate hazard and risk assessments of the importance of single cytokines in the context of biotherapeutic safety evaluations. While TNF-α, IL-12p40 and IL-1β are important cytokines for host resistance to *Mtb*, the overall low incidence of TB cases observed in clinical studies with cytokine-specific neutralizing antibodies suggests that susceptibility to reactivation of LTBI is determined by a combination of factors rather than the deficiency of just one cytokine. For example, a functional interdependence between IL-1β and TNF-α regulates TNF-α-dependent control of *Mtb* infection [70]. Contrasting some of our interpretations, and despite few case reports in the context of UST therapy, post-marketing surveillance data suggest that ANA and UST would not significantly increase the risk of LTBI reactivation [17,71]. However, these conclusions may be biased by the compounds' labels which advise to perform LTBI testing prior to initiating treatment and, if positive, provide anti-TB prophylaxis which has been shown to reduce drastically the advent of reactivation cases [72]. Our observations rather suggest that special considerations should be taken in future, should these biologics access markets of low and middle-income countries harboring higher TB incidence and where the use of such drugs could also increase the risk of direct progression to disease following a primary infection.

In conclusion, the data presented here demonstrate the clinical translational relevance and versatility of the human, *in-vitro* granuloma model by enabling mechanistic studies and allowing comparative profiling of the impact of specific immunological pathways in the context of

*Mtb* dormancy and resuscitation. Altogether our results support *in-vitro* granulomas as a valuable tool for preclinical evaluation of the risk of new biological therapies that could promote LTBI reactivation. Given the length of treatment and potential side effects of drugs used for TB preventive therapy [72], assessing preclinically this risk, and subsequent need for LTBI screening and prophylaxis, could eventually benefit clinical decision making and patient safety.

## Materials and methods

### Ethics statement

Human peripheral blood mononuclear cells were isolated from buffy coats obtained from the Interregionale Blutspende SKR AG, Bern, Switzerland. All donors provided informed consent which includes information on the use of blood products for research purposes (https://www.jedonnemonsang.ch/fileadmin/pdf_form/Informationsblatt_Spender_2019_f.pdf).

### Antibodies and reagents

Human IgG1 isotype control (Biolegend; clone ET901), anti-human TNF-α adalimumab (Humira, Abbvie), soluble human TNFR2-Fc fusion protein etanercept (Enbrel, AMGEN), anti-human IL-17A secukinumab (Cosentyx, Novartis) and anti-human IL-12p40 Ustekinumab (Stelara, Janssen) were used at a final concentration of 10 ng/ml unless specified otherwise in the figure. Recombinant IL-1RA anakinra (Kineret, Swedish Orphan Biovitrum) was used at 1.15 ng/ml (equimolar). TGF-β1-neutralizing antibody (Biolegend; clone 19D8) was used at 1 μg/ml. Other antibodies were obtained from Biolegend if not stated differently. IFN gamma, TNF alpha and TNF beta Human ProcartaPlex™ Simplex Kits (Invitrogen), Cytokine & Chemokine 34-Plex Human ProcartaPlex™ Panel 1A (Invitrogen) and Magnetic Luminex Performance Assay TGF-beta 1 Kit (R&D Systems) were used for cytokine quantification.

### Human peripheral blood mononuclear cells (PBMCs)

PBMCs were isolated by Ficoll-Paque (GE Healthcare) density-gradient centrifugation of buffy coats from healthy blood donors (Interregionale Blutspende SKR AG, Bern, Switzerland), as per written informed consent. After two washings in RPMI medium (Sigma), PBMC aliquots were cryopreserved in RPMI containing 10% DMSO (Sigma) and 40% fetal bovine serum (FBS, Gibco) and stored in liquid nitrogen until use. When needed, PBMCs were thawed, washed twice in RPMI containing 10% FBS and benzonase (12.5 U/ml, BioVision) and rested in RPMI containing 10% FBS for at least 6 h at 37°C (5% $CO_2$). Sample viability above 95% was assessed by trypan blue dye exclusion method and concentration adjusted to $10^7$ cells/ml in RPMI containing 20% human serum (PAN-Biotech) (referred to as "cell culture medium" from here on). To investigate the presence of CD4$^+$ T cell memory response against mycobacterial antigens, PBMCs were stimulated or not with ESAT-6/CFP-10/TB7.7 peptide pool (Peptides & elephants, 1 μg peptide/ml final) or PPD (Statens serum institute, RT23, 10 μg/ml final) in the presence of Brefeldin A (Biolegend). After overnight incubation, PBMCs were fixed and washed with intra-cellular staining buffers from Biolegend and stained with anti-human CD3-FITC (clone OKT3); anti-human CD4-FITC (clone RPA-T4); anti-human CD8a-APC (clone, HIT8a) and anti-human IFN-γ-PerCP (clone 4S.B3) before data acquisition on a BD FACSCalibur instrument and analysis using FlowJo10.5 (S1 Fig).

### Isolation of PBMC subsets and fluorescent staining

Rested PBMCs were sequentially subjected to CD14 and CD4 selection using magnetic microbeads (Miltenyi Biotec GmbH). Isolated monocytes and CD4$^+$ T cells were next stained

in orange and green, respectively, using Live Cell Tracking Kits (Abnova) as per manufacturer's instructions, while the unselected PBMC subsets were left unstained. Finally, PBMCs were reconstituted from the various fractions according to the proportion of CD14$^+$ and CD4$^+$ populations in the initial sample.

## *M. tuberculosis* (*Mtb*)

*Mtb* H37Rv was grown in 7H9 broth supplemented with 10% ADC (5% bovine albumin fraction V, 2% dextrose and 0.003% catalase), 0.5% glycerol (AppliChem Panreac) and 0.1% Tween-80 (Sigma) under gentle agitation to mid-exponential phase (OD$_{600}$ approximately 0.6). Bacteria were then washed with PBS containing 0.1% Tween-80, re-suspended in cell culture medium, water-bath sonicated for 2 min and centrifuged at 260×g for 5 min. The upper part of the supernatant (single-bacteria suspension) was recovered, cryopreserved by adding 5% glycerol (final) and stored at –80˚C until use. Concentration of the frozen stocks was quantified by colony forming units (CFU) assessment.

## 3D *in-vitro*, *Mtb*-induced human granulomas

The human, *in-vitro* granuloma model developed by Kolattukudy and colleagues [28] was adapted. Briefly, rested PBMCs were infected with *Mtb* at a multiplicity of infection (MOI) of 0.05 bacteria per monocyte, assuming 10% monocytes in PBMCs, and distributed in 24-well plates at 2.5×10$^6$ PBMCs/well. An extracellular matrix (ECM) was prepared by mixing thoroughly 0.95 ml of PureCol collagen solution (Advanced BioMatrix), 50 μl of 10×DPBS (SAFC Biosciences), 4 μl of fibronectin (Sigma), and 10 μl of 1N NaOH (Sigma) per ml of ECM required and kept at 4˚C. The ECM solution was mixed with the infected PBMCs in a 1:1 ratio (v/v) at room temperature (RT), and was allowed to set for 45 min at 37˚C (5% CO$_2$). Once the ECM completely set, wells were topped up with 500 μl of cell culture medium and incubated at 37˚C (5% CO$_2$). On day 4 post-infection, when relevant, supernatant was replaced by the same amount of fresh cell culture medium containing the studied antibodies, biologics or the isotype control. Granuloma formation was monitored on day 7–8 post-infection using a Leica DM IL LED inverted microscope and a Leica MC170 HD camera (Leica).

## Papain digestion and Fab-fragment purification

Fab fragments from adalimumab and an anti-LT-α antibody (clone 359-238-8) were generated and purified using Pierce$^{TM}$ Fab Micro Preparation Kit (ThermoFisher Scientific) as per manufacturer's instructions. Briefly, 50 μg of each antibody were diluted in Digestion Buffer and desalted using Zeba$^{TM}$ Spin Desalting Columns prior to digestion with immobilized papain for 6 h at 37˚C in an end-over-end mixer. The Fab fragments were then purified using NAb$^{TM}$ Protein A Plus Spin Columns. Protein concentration was determined by measuring absorbance at 280 nm (using an estimated extinction coefficient of 1.4) and purity of the isolated Fab fragments was confirmed in reducing SDS-PAGE and Coomasie blue staining (S3 Fig). When used for the treatment of *Mtb*-induced granulomas, concentration was normalized according to total TNF-α binding sites in adalimumab.

## Dual auramine-O/Nile red staining of *Mtb*

At the specified time-points, supernatant was removed and wells were treated with 250 μl of collagenase (1 mg/ml; Sigma) for 40 min at 37˚C (5% CO$_2$) to release the PBMCs from the ECM. Host cells were pelleted at 400×g for 5 min and lysed with 0.1% Triton X-100 (Sigma) for 20 min at RT, followed by centrifugation at 6000×g for 5 min to obtain the *Mtb* pellet.

Bacilli were inactivated with 1× CellFIX (BD) for 20 min at RT. Fixed samples were put on glass slides, air dried and heat fixed at 70˚C. Fluorescent acid-fast staining using TB Fluorescent Stain Kit M (BD) was performed in combination with neutral-lipid staining dye Nile red (Sigma) [28]. Each sample was stained with auramine-O for 20 min, decolorized for 30 s, covered with Nile red (10 μg/ml) for 15 min and counterstained with potassium permanganate for 2 min, including gentle washes with distilled water between each step. Air-dried, stained slides were mounted using Vectashield mounting medium (Vectorlabs) and examined using a Leica DM5000 B fluorescence microscope (Leica). For quantification purposes, at least 200 bacteria per sample were counted. Representative micrographs for auramine-O and Nile red-positive *Mtb* are shown in S4 Fig.

## Rifampicin (Rif) tolerance and CFU assessment

To evaluate Rif tolerance at day 8 post-infection, *Mtb*-infected PBMC exposed to 10 ng/ml of adalimumab, etanercept or an isotype control for 4 days were either left untreated (control) or treated with 5 μg/ml Rif and incubated at 37˚C (5% $CO_2$) for 3 additional days as described previously [28]. Then, *Mtb* was recovered after collagenase treatment and PBMC lysis and the pellet was re-suspended in 1 ml of $H_2O$ containing 0.05% tween-80. To determine the number of CFU, 10-fold serial dilutions were prepared in triplicate in PBS containing 0.1% tween-80 and plated on Middlebrook 7H11 agar plates supplemented with 0.5% glycerol and 10% OADC (0.05% oleic acid in ADC). The percentage of Rif tolerance was calculated by using the formula: %Rif tolerance = CFU(Rif-treated)/CFU(untreated)×100.

## RNA isolation and gene expression analysis by qRT-PCR

At day 8 post-infection, *Mtb* was recovered from *in-vitro* granulomas after collagenase treatment and Triton X-100 lysis. The bacterial pellet was resuspended in TRI-reagent (Zymo Research) and stored at –80˚C until RNA extraction. The TRI-reagent suspension was transferred into Beadbug tubes containing 0.1 mm silica beads (Sigma) and bead-beaten for 45 s at 6.5 m/s using a FastPrep-24 (MP Biomedicals). After removing the cellular debris by centrifugation, total RNA was extracted from the bacterial lysate using Direct-zol RNA MiniPrep (Zymo Research) according to the manufacturer's protocol. Total RNA was treated with DNAse I (Invitrogen), purified using RNeasy MinElute Cleanup Kit (Qiagen) and reverse transcribed using FIREScript RT cDNA Synthesis Mix (Solis Biodyne) as per manufacturer's instructions. Expression of *icl1*, *gltA1*, *nuoB*, *ctaD* and 16S rRNA were quantified by qPCR using Hot FIREpol EvaGreen qPCR Mix Plus (ROX) (Solis Biodyne) in a StepOnePlus real-time system (Applied Biosystems). The details (sequences and references) of the primers used are provided in S1 Table.

## Cytokine measurements

Supernatants were collected at the specified time-points and stored at –80˚C until filter-sterilization and analysis within 24 h. Concentrations of selected cytokines were determined using magnetic-bead arrays (see Antibodies and Reagents) on a Luminex Bio-Plex 200 platform and Bio-Plex Manager 6.0 software (Bio-Rad) according to the manufacturer's recommendations.

## Flow cytometry analysis

Host cells were recovered from the ECM after an overnight incubation with brefeldin A (Biolegend) and collagenase treatment at the indicated time-points and pelleted at 400×g for 5 min as described above. Cells were then stained with anti-human CD40-FITC (clone 5C3), anti-

human CD206-PE (clone 15–2), anti-human TNF-α-PerCP (clone MAb11) and anti-human HLA-DR-APC (clone L243) (macrophages) or anti-human CD3-FITC (clone OKT3), anti-human CD4-FITC (clone RPA-TA), anti-human LT-α-PE (clone 359-81-11), anti-human TNF-α-PerCP (clone MAb11) and anti-human CD8α-APC (clone HIT8a) (T cells) following a standard protocol. Briefly, cells were incubated in 50 μl of PBS containing 1% FBS and 1 μl of each of the antibodies against extracellular markers for 20 min at RT. Samples were washed once with PBS containing 1% FBS, fixed in Fixation buffer (Biolegend) for 20 min at RT and washed twice with 1× Intracellular staining permeabilization wash buffer (ICS perm/wash buffer, Biolegend). Cells were then incubated in 50 μl of 1× ICS perm/wash buffer containing 1 μl of each of the antibodies against intracellular markers for 30 min at RT, washed once with 1× ICS perm/wash buffer and fixed in 1× CellFIX for 20 min at RT. At least 50,000 events per sample were acquired on a BD FACSCalibur™ instrument using CellQuest Pro software (BD) and processed using FlowJo 10.5.

## Quantification and statistical analysis

GraphPad Prism 7 or R.3.5.1 and R studio 1.1.456 were used to generate quantitative graphical representation of the generated data and statistical tests. The number of independent donors used (biological replicates), nature of the tests and definition of center and dispersion measures is specified within the respective figure legend. A single technical replicate per donor and condition was tested to generate each figure; with the exceptions of Fig 3B were up to five technical replicates were cumulated depending on the donor, and Fig 3C were all the conditions were tested in duplicates. For all figures significance was defined as: n.s., not significant; *, $p<0.05$, **, $p<0.01$; ***, $p<0.001$; ****, $p<0.0001$.

## Supporting information

**S1 Fig. Immunization status of the blood donors selected for the study.** PBMCs were stimulated overnight with *Mtb* protein purified derivative (PPD) or a synthetic peptide pool from ESAT-6, CFP-10 and TB7.7 *Mtb* proteins and analyzed by flow cytometry. **(A)** Representative dotplots of the gating strategy. **(B)** Background-subtracted frequencies of IFN-γ-producing CD4+ T cells for each donor selected for the study. The response was considered positive when more than 0.05% of cytokine-producing cells were detected within the CD4+ T cell parent population and this frequency was at least twice higher than the background level detected in the absence of stimuli.
(TIF)

**S2 Fig. Gating strategy used to detect TNF-α- or LT-α-producing cell types within *Mtb*-induced granuloma.** Representative dotplots showing the gating strategy used to focus on HLA-DR+ macrophage **(A)** or CD4+ and CD8+ T cell populations **(B)**.
(TIF)

**S3 Fig. SDS-PAGE analysis of undigested and papain-digested Fab fragments from ADA and anti-LT-α antibodies.** SDS-PAGE and Coomasie blue staining of adalimumab (ADA) **(A)** or an anti-LT-α antibody **(B)** and their purified Fab fragments (ADA-Fab and α-LT-α-Fab, respectively).
(TIF)

**S4 Fig. Representative micrographs for auramine-O and Nile red staining.** *Mtb* H37Rv recovered from granulomas 8 days post-infection and after 4 days of exposure to either adalimumab, etanercept or an isotype control.
(TIF)

**S1 Table. Sequences and references of the primers used for qPCR.**
(PDF)

**S2 Table. Statistical analysis of Fig 5C and 5D.** Statistical analysis was performed using a generalized linear mixed-effects model; n.s., not significant; *, p<0.05, **, p<0.01; ***, p<0.001; ****, p<0.0001.
(PDF)

**S1 Data. Raw data used to generate the figures on this manuscript.** Each row contains the values from one independent donor.
(XLSX)

## Acknowledgments

We would like to thank Christian Schindler for statistical support and script writing of the R code used for generalized linear mixed-effects model.

## Author Contributions

**Conceptualization:** Dominique Brees, Salah-Dine Chibout, Todd Fox, Michael Kammüller, Damien Portevin.

**Data curation:** Ainhoa Arbués.

**Formal analysis:** Ainhoa Arbués, Damien Portevin.

**Investigation:** Ainhoa Arbués, Damien Portevin.

**Methodology:** Ainhoa Arbués, Damien Portevin.

**Project administration:** Damien Portevin.

**Supervision:** Michael Kammüller, Damien Portevin.

**Writing – original draft:** Ainhoa Arbués, Damien Portevin.

**Writing – review & editing:** Ainhoa Arbués, Dominique Brees, Salah-Dine Chibout, Todd Fox, Michael Kammüller, Damien Portevin.

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
