## [Decision Letter · Decision Letter 0]

28 Aug 2019

Dear PhD Portevin,

Thank you very much for submitting your manuscript "TNF-α antagonists differentially induce TGF-β1-dependent Mycobacterium tuberculosis reactivation" (PPATHOGENS-D-19-01320) for review by PLOS Pathogens. Your manuscript was fully evaluated at the editorial level and by independent peer reviewers. The reviewers appreciated the attention to an important problem, but raised some substantial concerns about the manuscript as it currently stands. These issues must be addressed before we would be willing to consider a revised version of your study. We cannot, of course, promise publication at that time.

We therefore ask you to modify the manuscript according to the review recommendations before we can consider your manuscript for acceptance. Your revisions should address the specific points made by each reviewer.

(1) A letter containing a detailed list of your responses to the review comments and a description of the changes you have made in the manuscript. Please note while forming your response, if your article is accepted, you may have the opportunity to make the peer review history publicly available. The record will include editor decision letters (with reviews) and your responses to reviewer comments. If eligible, we will contact you to opt in or out.

(2) Two versions of the manuscript: one with either highlights or tracked changes denoting where the text has been changed; the other a clean version (uploaded as the manuscript file).

Additionally, to enhance the reproducibility of your results, PLOS recommends that you deposit your laboratory protocols in protocols.io, where a protocol can be assigned its own identifier (DOI) such that it can be cited independently in the future. For instructions see http://journals.plos.org/plospathogens/s/submission-guidelines#loc-materials-and-methods

We hope to receive your revised manuscript within 60 days. If you anticipate any delay in its return, we ask that you let us know the expected resubmission date by replying to this email. Revised manuscripts received beyond 60 days may require evaluation and peer review similar to that applied to newly submitted manuscripts.

[LINK]

Sincerely,

Thomas R. Hawn

Associate Editor

PLOS Pathogens

Sabine Ehrt

Section Editor

PLOS Pathogens

Kasturi Haldar

Editor-in-Chief

PLOS Pathogens

orcid.org/0000-0001-5065-158X

Grant McFadden

Editor-in-Chief

PLOS Pathogens

orcid.org/0000-0002-2556-3526

Reviewer's Responses to Questions

**Part I - Summary**

Reviewer #1: The author reports use of a culture system for aggregates of PBMC described as “in vitro granulomas” to investigate the differential risk of TB progression with several disease modifying antirheumatic drugs. The study addresses a clinically relevant gap in knowledge and the results for the most part parallel differences identified in clinical studies, particularly for ADA vs ETA. The work represents a novel use of the culture system for comparative studies of different biological agents and provides evidence for a TB reactivation pathway mediated by TGF-beta following crosslinking of tmTNF-alpha by ADA, as well as a previously unsuspected role for LT- alpha in maintaining latency. The author contends that the in vitro culture system provides an efficient model to assess TB risk of immune-modulating biological therapies, although the mechanism of the putative dormancy that is present in >20% of bacilli at day 1 and increases to >40% by day 8 in untreated cultures is unclear and might not reflect the mechanisms that enforce latency in vivo.

While the execution of experiments appears to have been robust, the author relies heavily on differential staining with Auramine O and Nile red to discriminate between metabolically active and “dormant” bacilli. As note in Part II, it can be argued that this approach lacks stringency. Furthermore, while the trends for changes in the proportions Auramine O and Nile red bacilli following treatment with ADA, ETA, UST and ANA were statistically significant and to a certain extent match the clinical data, the small magnitude of absolute differences between these agents in this assay raise questions about biological significance. This concern is even greater for the measure of relative rifampin resistance between ADA and ETA.

Reviewer #2: The work from Arbues et al. builds on a model in which human PBMCs are assembled into granuloma-like structures, previously described in 2013 from Kalattakudy's group to examine the effect of various biologics targeting TNF. They carry out many of the same experiments conceptually as the 2013 paper, including monitoring of auramine/nile red ratio, as a measure of metabolic activity of the bacteria. They then go on to test a number of biologics that are used clinically and reproduce a difference in the effects of two drugs that depends on the mechanism of action. This is a clever use of these assays, the experiments are generally well carried out, and they seem to have uncovered some differences in mechanism and the interventions. My major concern is that almost everything relies on a single assay (auramine/nile red ratio) which the authors define somewhat arbitrarily as latency/dormancy. At minimum, there needs to be additional support for the claims around dormancy as well as consideration (cfu, bacterial numbers) of overall effects on CFU in this model.

Many of the differences shown (e.g. TGFbeta) are quite small in terms of the overall changes in ratio so, while interesting, it feels a bit dangerous to extrapolate from a single ratio-based readout in a single technical approach.

Reviewer #3: This manuscript by Arbues et al., explores the mechanism of TNF antagonist-mediated Mycobacterium tuberculosis (Mtb) reactivation. Using an in vitro granuloma model, they test a panel of immunomodulators currently in clinical use to determine their effects on Mtb reactivation in this model system. They show that two TNF-neutralizing biologics showed differential effects in their system. Adalimumab promoted Mtb Auramine-O staining, which is interpreted as a readout for active Mtb metabolism/replication, whereas etanercept induced more Mtb Nile red staining, which is taken as indicative of dormancy. These observations, as interpreted, parallel the clinical scenario, in which Adalimumab treatment results in a higher risk of Mtb reactivaton than Etanercept treatment. They went on to show that these effects were not mediated by TNF neutralization, but rather, were due to reverse signaling through membrane TNF, by Adalimumab (which is bivalent), but not by Etanercept (which is monovalent) and that this signaling induced TGFb, which in turn was responsible for the reactivation. This is a novel idea, which if true, would be an important finding for the field. However, the data presented, relying exclusively on these staining approaches without showing validation that these staining patterns represent increased replication or bacterial loads, is not sufficient to convincingly support their findings. More experimental evidence is needed to bolster their conclusions.

Reviewer #4: This study uses an in vitro model of human T cell activation by very low dose M. tuberculosis (M.tb) infection of PBMC within an extracellular matrix of collagen and fibronectin to analyse the effect of different anti-cytokine “biologics” on “reactivation” of M.tb as measured by the ratio of acid-fast and lipid containing mycobacteria. They infer this is equivalent of chronic granulomas in humans with latent TB infection (LTBI), but this is an overstatement of the significance of the model. Nevertheless, they make interesting and novel observations on the differential capacity of different anti-cytokine therapies to increase the recovery of acid-fast M.tb that they infer are metabolically active. They propose that reverse signalling through memTNF receptors on macrophages by bivalent anti-TNF mAb (ADA) stimulates TGF-b production that contributes to the increased potential of ADA to cause “reactivation” compared to soluble TNFR2 inhibitor (ETA). This is supported by effect of anti-TGF-b antibodies, but there are some inconsistencies in the data that should be addressed. They also highlight the effect of another TNFSF member, LT-a, in preventing reactivation.

Issues:

1. Model:

The cellular aggregates of T cells and macrophages are followed for 8 days only and do not have all the hallmarks of chronic granulomas in humans with LTBI, eg epithelioid macrophages, giant cells and fibrosis from mesenchymal cells. Therefore, although a useful model, it is not equivalent to chronic granulomas, and this should be acknowledged and discussed. The “reactivation” is based on ratio of acid-fast mycobacteria to those containing lipid bodies over 8 days. Have they or others demonstrated differences in activation of dormancy genes in this model? Does the addition of anti-TNF mAb lead to increased numbers of bacteria cultured from the cellular aggregates? Results from supporting mycobacterial cultures would strengthen the findings from the model.

2. Selection of subjects:

These appeared to be from blood donors selected because of IFNg-secreting T cell response to PPD, but M.tb-specific T cell antigens. Does this mean they were BCG-vaccinated and not M.tb infected? Why were these referred to as ”pre-immune” donors (L313) when they were PPD-positive? Have they or others compared responses in well characterised subjects with definite LTBI or active TB? What is the response in this model from uninfected controls, ie PPD-negative subjects.

3. Results

- The effect of ADA in system was reproducible in different experiments, but ETA had stronger effect on “reactivation” in some experiments eg Fig 1C than in others, eg Fig 2, 4 and 5. Although anti-IL-17 had no effect in vitro, it may have an effect on granulomas within the lungs, and this difference should be recognised.

- The number of donors in each figures should be clarified, eg Fig 2C, are biological replicates the same as donors or multiple samples from the same donor; and how many technical replicates were tested?

- Fig 2 D: the ADA and ETA samples are not labelled. Were these the results from only 2 independent replicates and if so the SEM cannot be calculated?

- The levels of TGF-b measured (from only 4 donors) were small and there was no difference between ADA and ETA samples that does not support their hypothesis (Fig 4A), although exogenous TGF-b at higher concentrations did have an effect.

- The effect of anti-LTa in their model (Fig 5) does support a role for T-cell derived LTa on control of M.tb infection. But they should explain the statement about the “neutralization of one TNFR2 ligands somewhat interferes with the sensing of the other”(L297).

4. Statistical analysis

Different statistical methods were used in analysis of different results. Why was a generalized liner mixed-effects model used rather than a simpler non-parametric comparison between groups in some and not other experiments?

**Part II – Major Issues: Key Experiments Required for Acceptance**

Reviewer #1: Differential staining with Auramine O and Nile red is used to discriminate between biologically active vs dormant Mtb, but it is unclear whether conclusions regarding metabolism can be reliably drawn. Non-replicating persistence induced through oxygen starvation results in Mtb cells that have lipid bodies that stain with Nile red, but this can occur in a variety of conditions and might not accurately report cellular metabolic status. Similar considerations apply to Auramine O staining, which was reported to be unaffected by antimicrobial treatment (Kamariza et al. Sci Translat Med 10[430]: eaam6310). The author might consider additional tests to demonstrate dormancy, such as inability to grow on agar but retention of the capacity to grow in permissive liquid media. Another option would be Mtb gene expression profiling, as reported to Kapoor et al. (PLoS One 8[1]:e53657) who were the first to show that neutralizing TNF-alpha reduced the proportion of bacilli with features of dormancy in the in vitro granuloma model.

Only PBMC samples containing CD4+ T cells that made IFN-gamma in response to PPD were used for these experiments, yet the author states that none of the donors displayed signs of LTBI based on the absence of response to immunodominant Mtb peptides. What is the basis for IFN-gamma production by CD4+ T cells from TB-naïve donors? Since the culture system meant to model immunologically enforced dormancy as occurs in LTBI, why not use donors with LTBI. Differences in cultures from donors with or without LTBI might be informative.

Despite case reports of TB progression following treatment with UST, the evidence that this is a potent effect in humans is weak (Cantini et al. Mediators of Inflamm 2017:8909834). The finding of high activity of UST in the Auramine O/Nile red assay despite very low levels of IL-12p70 and IL-23 in the cultures is mechanistically unexplained and raises some concern about the validity of the model, at least for testing this particular biological agent.

Reviewer #2: As I wrote above, the extrapolation to latency/dormancy/reactivation in the time frame and ratio-based readout of the experiments seems a bit of a stretch.

1) There needs to be some investigation and validation of what the nile red/auramine populations are in terms of dormancy/latency. What is the justification for calling nile red positive bacteria truly dormant? It is also strange to me (see point below) that everything is graphed as ratios. While there may be some justification for this in terms of the complexity/scoring of the assay, it seems to me important to know what the absolute numbers of bacilli are with each treatment.

2) I think there needs to be some cfu data presented along with the ratio readouts to go along with these assays. The ratio is a useful metric, but could mean quite different things depending on what happens to overall numbers of bacteria in each condition.

Reviewer #3: 1) Prior work by other groups working with the in vitro granuloma model have used immunofluorescent antibody staining to show that the immune aggregates formed in vitro bear some features of in vivo granulomas. However, in this paper the authors show only a bright field aggregate of cells. The authors should show by immunofluorescence that these aggregrates, in their own hands, show organizational features that resemble granulomas.

2) The manuscript’s conclusions depend entirely on the postulate that Auramine-O staining indicates metablically active/replicating bacteria whereas Nile red staining represents dormant bacteria, however, no experimental evidence is provided to validate these assumptions. This is essential. In addition, bacterial load determinations should also be shown. If Mtb has reactivated in Adalimumab-treated granulomas, the CFUs should be higher.

3) Fig. 5 depicts the effects of lymphotoxin neutralization on auromine-O and Nile red staining. These experiments seem only loosely related, at best, to the studies of the TNF antagonists and their differential impact of TGFb production via membrane TNF signaling. It is unclear what value these data add to this manuscript. The manuscript would be strengthened by bolstering the experiments regarding the TNF antagonists and their differential impact on TGFb and reactivation (as discussed above) and by eliminating these LT experiments (which are unrelated, and less developed). As is, seems like an odd way to end the paper.

4) In the Introduction (lines 88-96) discuss previous experimental studies that attempt to define the mechanism that could be responsible for the differential risk of reactivation observed between anti-TNF antibodies and the receptor fusion protein, however, the authors overlook an intriguing study be Steffen Stenger’s group which may be relevant to their own findings (PMID: 19381021). In this study, the author’s observed lower numbers of effector T cells that expressed effector molecules (perforin and granulysin) in individuals treated with anti-TNF antibodies compared to those treated with receptor fusion proteins. They showed that antibodies could mediated complement-mediated lysis of effector T cells that expressed surface TNF, whereas the receptor fusion proteins could not, and suggested that this could explain the differential effect on reactivation (if effector T cells were critical for holding in check). However, the reduction in effector T cells could alternatively be due to intrinsic TGFb produced by reverse membrane TNF signaling. This seems worthy to include both in the Introduction and the Discussion if the findings can be validated and shown convincingly.

Reviewer #4: 1. Does the addition of anti-TNF mAb lead to increased numbers of bacteria cultured from the cellular aggregates? Results from supporting mycobacterial cultures would strengthen the findings from the model.

2. The levels of TGF-b in ADA and ETA inhibited samples were measured in only 4 donors and were small. There was no difference between ADA and ETA samples that does not support their hypothesis (Fig 4A). This should be repeated from more samples.

**Part III – Minor Issues: Editorial and Data Presentation Modifications**

Reviewer #1: The seven paragraph Introduction and the Discussion could be trimmed without loss of information. The author should work to avoid redundancies between these two sections but should also consider an expanded discussion of potential weaknesses in vitro culture system as a model of granulomas forming in vivo.

The author might consider a supplemental figure with micrographs to allay any concerns about the accurate discrimination between Auramine O vs Nile red positive bacterial cells. If there is any ambiguity in staining, then it would be necessary for the population counts to be performed by a blinded reader.

Reviewer #2: I think the authors need to be more precise in defining what they have shown. There have been long discussions about what constitutes latency and dormancy and reactivation in tuberculosis, and the simplified view throughout the paper is that dormancy simply corresponds to a nile red postive population. The writing should be adjusted accordingly throughout and, as mentioned above, these bacterial populations should be probed in some other way experimentally.

How do these data correspond with genetic manipulations of the TNF axis? Are there genetic manipulations that could be achieved in this model that would corroborate the findings from the biologics targeting the TNF pathway, particularly in regards to TGFb, which is an interesting finding but with rather mild effects in this model.

Reviewer #3: 1) In the first paragraph of the introduction, the authors go part way in their discussion of their discussion of “LTBI”. However, I think it is important that the field discuss the definition of LTBI more directly. I suggest something like, “its estimated that one quarter of the world has immunological evidence of past infection with Mtb. Although this population is frequently said to have LTBI, the % that truly harbor Mtb is not clear. These individuals likely have a range of outcomes of infection, including cure or eradication, etc. etc.

2) The sentence in lines 51-53 is likely inaccurate (5-10% progression from latent to active when immune system is weakened). Because most of these cases occur in regions where Mtb transmission is frequent, and several studies in these setting suggest that most of these are not reactivation events, but represent recent infections.

3) In discussing Fig 2A, the authors conclude that TNF production is bi-modal with one peak at day 1 post-infection and the other at day 8. However, no statistics are provided and the differences do not appear significant. This analysis needs to be bolstered with statistics, or the conclusions should be softened.

4) The difference in Rif-tolerant Mtb between ADA and ETA may be statistically significant (Fig. 2C), but are very, very modest, and of questionable biologic importance. These conclusions should be softened.

5) Figure 2D is very difficult to interpret because the various lines on the graph are not labeled or defined in the figure legend. Same comment for Figure 4B.

6) The difference between Iso and ADA in Fig. 4A does not appear statistically significant. The author’s report that it is significant using Friedman’s test? Is this the most appropriate test to use here? Is this a paired analysis? If so, should the data points from a single patient be connected by lines or color-coded?

7) In line 312, “we present a tissue of evidence . . . “ Is this a typo?

Reviewer #4: (No Response)

PLOS authors have the option to publish the peer review history of their article (what does this mean?). If published, this will include your full peer review and any attached files.

Reviewer #1: No

Reviewer #2: No

Reviewer #3: No

Reviewer #4: No

---

## [Editor Report · Decision Letter 1]

8 Jan 2020

Dear PhD Portevin,

We are pleased to inform that your manuscript, "TNF-α antagonists differentially induce TGF-β1-dependent resuscitation of dormant-like Mycobacterium tuberculosis", has been editorially accepted for publication at PLOS Pathogens. 

Before your manuscript can be formally accepted and sent to production, you will need to complete our formatting changes, which you will receive by email within a week. Please note that your manuscript will not be scheduled for publication until you have made the required changes.

IMPORTANT NOTES

(1) Please note, once your paper is accepted, an uncorrected proof of your manuscript will be published online ahead of the final version, unless you’ve already opted out via the online submission form. If, for any reason, you do not want an earlier version of your manuscript published online or are unsure if you have already indicated as such, please let the journal staff know immediately at plospathogens@plos.org.

(2) Copyediting and Proofreading: The corresponding author will receive a typeset proof for review, to ensure errors have not been introduced during production. Please review the PDF proof of your manuscript carefully, as this is the last chance to correct any errors. Please note that major changes, or those which affect the scientific understanding of the work, will likely cause delays to the publication date of your manuscript. 

(3) Appropriate Figure Files: Please remove all name and figure # text from your figure files. Please also take this time to check that your figures are of high resolution, which will improve the readbility of your figures and help expedite your manuscript's publication. Please note that figures must have been originally created at 300dpi or higher. Do not manually increase the resolution of your files. For instructions on how to properly obtain high quality images, please review our Figure Guidelines, with examples at: http://journals.plos.org/plospathogens/s/figures.

(4) Striking Image: Please upload a striking still image to accompany your article if one is available (you can include a new image or an existing one from within your manuscript). Should your paper be accepted, this image will be considered for our monthly issue image and may also appear on our website to feature your article. Please upload this as a separate file, selecting "striking image" as the file type upon upload. Please also include a separate "Other" file with a caption, including credits and any potential copyright information. Please do not include the caption in the main article file. If your image is from someone other than yourself, please ensure that the artist has read and agreed to the terms and conditions of the Creative Commons Attribution License at http://journals.plos.org/plospathogens/s/content-license. Please note that PLOS cannot publish copyrighted images.

(5) Press Release or Related Media: If your institution or institutions have a press office, please notify them about your upcoming paper at this point, to enable them to help maximize its impact. If they will be preparing press materials for this manuscript, please inform our press team in advance at plospathogens@plos.org as soon as possible. We ask that you contact us within one week to plan ahead of our fast Production schedule. If you need to know your paper's publication date for related media purposes, you must coordinate with our press team, and your manuscript will remain under a strict press embargo until the publication date and time. This means an early version of your manuscript will not be published ahead of your final version. 

(6)  PLOS requires an ORCID iD for all corresponding authors on papers submitted after December 6th, 2016. Please ensure that you have an ORCID iD and that it is validated in Editorial Manager.  To do this, go to ‘Update my Information’ (in the upper left-hand corner of the main menu), and click on the Fetch/Validate link next to the ORCID field.  This will take you to the ORCID site and allow you to create a new iD or authenticate a pre-existing iD in Editorial Manager

(7) Update your Profile Information: Now that your manuscript has been provisionally accepted, please log into Editorial Manager and update your profile, if needed. Go to https://www.editorialmanager.com/ppathogens, log in, and click on the "Update My Information" link at the top of the page. Please update your user information to ensure an efficient production and billing process. 

(8) LaTeX users only: Our staff will ask you to upload a TEX file in addition to the PDF before the paper can be sent to typesetting, so please carefully review our Latex Guidelines http://journals.plos.org/plospathogens/s/latex in the meantime.

(9) If you have associated protocols in protocols.io, please ensure that you make them public before publication to guarantee immediate access to the methodological details.

Best regards,

Thomas R. Hawn

Associate Editor

PLOS Pathogens

Sabine Ehrt

Section Editor

PLOS Pathogens

Kasturi Haldar

Editor-in-Chief

PLOS Pathogens

orcid.org/0000-0001-5065-158X

Michael Malim

Editor-in-Chief

PLOS Pathogens

orcid.org/0000-0002-7699-2064
---

## [Editor Report · Acceptance letter]

10 Feb 2020

Dear PhD Portevin,

We are delighted to inform you that your manuscript, "TNF-α antagonists differentially induce TGF-β1-dependent resuscitation of dormant-like *Mycobacterium tuberculosis*," has been formally accepted for publication in PLOS Pathogens.

Best regards,

Kasturi Haldar

Editor-in-Chief

PLOS Pathogens

orcid.org/0000-0001-5065-158X

Michael Malim

Editor-in-Chief

PLOS Pathogens

orcid.org/0000-0002-7699-2064